# Investigation of Vestibular Function in Adult Patients with Gitelman Syndrome: Results of an Observational Study

**DOI:** 10.3390/jcm9113790

**Published:** 2020-11-23

**Authors:** Mihaela Alexandru, Marie Courbebaisse, Christine Le Pajolec, Adeline Ménage, Jean-François Papon, Rosa Vargas-Poussou, Jérôme Nevoux, Anne Blanchard

**Affiliations:** 1AP-HP, Université Paris Saclay, Hôpital Bicêtre, Service d’Oto-Rhino-Laryngologie, 94270 Le Kremlin-Bicêtre, France; mihaeladana.alexandru@aphp.fr (M.A.); christine.lepajolec@aphp.fr (C.L.P.); adeline.menage@aphp.fr (A.M.); jean-francois.papon@aphp.fr (J.-F.P.); 2AP-HP, Centre—Université de Paris, Hôpital Européen Georges-Pompidou, Service de Physiologie-Exploration Fonctionnelles Rénales, 75015 Paris, France; marie.courbebaisse@aphp.fr; 3Faculté de Médecine Paris Descartes, Université de Paris, 75006 Paris, France; 4INSERM, U1151-CNRS UMR8253, 75015 Paris, France; 5Faculté de Médecine, Université Paris-Saclay, F-94275 Le Kremlin-Bicêtre, France; 6AP-HP, Centre—Université de Paris, Hôpital Européen Georges-Pompidou, Département de Génétique et Biologie Moléculaire, 75015 Paris, France; rosa.vargas@aphp.fr; 7INSERM, UMRS 1138, Centre de Recherche des Cordeliers, 75006 Paris, France; 8Centre de Référence des Maladies Rénales Héréditaires de l’Enfant et de l’Adulte (MARHEA), Hôpital Européen Georges-Pompidou, 75015 Paris, France; 9INSERM, U1120, Institut Pasteur, 75724 Paris CEDEX 15, France; 10AP-HP, Hôpital Européen Georges-Pompidou, Centre d’Investigation Clinique 1418, 75015 Paris, France

**Keywords:** Gitelman syndrome, vertigo, SLC12A3, NCC, endolymphatic sac, inner ear

## Abstract

Gitelman syndrome (GS) is a rare salt-losing tubulopathy caused by an inactivating mutation in the *SLC12A3* gene, encoding the thiazide-sensitive sodium chloride cotransporter (NCC). Patients with GS frequently complain of vertigo, usually attributed to hypovolemia. Because NCC is also located in the endolymphatic sac, we hypothesized that patients with GS might have vestibular dysfunction. Between April 2013 and September 2016, 20 (22%) out of 90 patients followed at the reference center complained of vertigo in the absence of orthostatic hypotension. Sixteen of them were referred to an otology department for investigation of vestibular function. The vertigo was of short duration and triggered in half of them by head rotation. Seven patients (44%) had a vestibular syndrome. Vestibular syndrome was defined: (1) clinically, as nystagmus triggered by the head shaking test (*n* = 5); and/or (2) paraclinically, as an abnormal video head impulse test (*n* = 0), abnormal kinetic test (*n* = 4) and/or abnormal bithermal caloric test (*n* = 3). Five patients had associated auditory signs (tinnitus, aural fullness or hearing loss). In conclusion, we found a high frequency of vestibular disorder in GS patients suffering from vertigo, suggesting a role of NCC in the inner ear. Referent physicians of these patients should be aware of this extrarenal manifestation that requires specific investigations and treatment.

## 1. Introduction

Gitelman syndrome (GS) (Online Mendelian Inheritance in Man No. 263800) is a rare nephrological disease characterized by hypokaliemia, hypomagnesemia, and hypocalciuria that mimics the effects of high doses of thiazide diuretics. The prevalence of the disease is 1 to 10 per 40,000 [1,2]. The disease is caused, in most cases, by an inactivating mutation that is recessively inherited in the *SLC12A3* gene, which encodes the sodium chloride cotransporter (NCC) [3]. The clinical and biological symptoms of GS include asthenia, cramps, muscle weakness, nycturia, low blood pressure and, in children, growth retardation. The most frequent extrarenal complication is chondrocalcinosis [4]. Between 20 and 50% of GS patients also complain of dizziness, usually attributed to hypotension or vagal sickness [4,5]. In our clinical practice, we observed that vertigo characteristics were suggestive of vestibular dysfunction, leading us to hypothesize a link between vertigo and GS. Although NCC was originally assumed to be selectively expressed in the kidney, it is also expressed in the bone [6] and in the medial part of the endolymphatic sac in the inner ear [7,8]. The endolymphatic sac is known to regulate endolymph homeostasis through an ion and water exchange (Appendix A), and NCC is thought to contribute to this homeostasis [7,8]. We hypothesized that defects in NCC in GS patients lead to modifications of endolymph regulation. The aim of our study was to identify vestibular defects in adults with GS. We presented a retrospective analysis of a self-evaluation of the symptoms and a vestibular function evaluation in 16 patients with vertigo not attributed to hypotension or vagal sickness.

## 2. Patients and Methods

### 2.1. Study Population

All adult patients enrolled in the present study belong to the cohort of GS patients followed at the Centre de référence des maladies rénales héréditaires de l’enfant et de l’adulte (MARHEA) in Georges Pompidou European Hospital, Paris, France. Data were extracted from the follow-up database for adult GS patients of the French congenital tubulopathies network. This cohort follow-up received the approval of the local ethics committee, the Comité de Protection des Personnes, Paris–Île de France XI (Ref. 09069).

From September 2016 to April 2019, as part of the routine care, 90 adults with GS were asked if they suffer from vertigo, defined as the sensation of self-motion (of head/body) when no self-motion occurs or the sensation of distorted self-motion during an otherwise normal head movement [9]. Patients with orthostatic hypotension were excluded. Orthostatic hypotension was defined as a decrease in systolic blood pressure of at least 20 mmHg and/or a decrease in diastolic blood pressure of 10 mmHg or more [10].

Patients suffering from vertigo were referred to the tertiary referral center for neuro-otological diseases at Bicêtre Hospital (Paris, France) for analysis of vestibular function. They were asked to answer seven questions to clarify symptoms, duration of the sensation, associated signs and triggers and if a specialist was consulted, a diagnosis made or treatment prescribed (Appendix A). All the patients referred to the tertiary center had genetically proven Gitelman syndrome (Appendix A [1,3,11,12,13,14,15,16,17,18,19,20,21,22]). None of these patients had a history of ear surgery, temporal bone fracture or tympanic perforation, infection or malignancies. No women were pregnant. Clinical data, including blood pressure measured at the time of medical consultation in sitting and standing positions, as well as the results of the most recent blood sample (i.e., within 3 months), were retrospectively extracted from medical records.

### 2.2. Vestibular Evaluations

#### 2.2.1. Neuro-Otological Clinical Exam

A complete neuro-otological exam was performed by an ear, nose and throat specialist. The examination included a complete neurological test, especially for evaluation of the cranial nerves and cerebellar functions, to determine whether a central cause of vertigo was present. Patients were evaluated for tinnitus and either aural fullness or hearing loss during vertigo crises. When associated with vertigo crises lasting at least 20 min, these symptoms correspond to the clinical triad of Menière’s disease [23]. Subjects were also asked if they had headache during their vertigo crises to evaluate the possibility of vestibular migraine [24]. To evaluate whether benign paroxysmal positional vertigo was present, the Dix–Hallpike maneuver was performed. The Fukuda stepping test was performed to evaluate the vestibulospinal reflex, which modulates the postural adjustments to head movements and position [25]. For this test, the patient was asked to close both eyes, extend both arms and walk in place for 50 steps. The test was considered abnormal if there was a postural deviation greater than 45°. The head shaking test (HST) was performed to evaluate the vestibulo-ocular reflex, which represents the control of eye movements by the vestibule. After the stimulation, the examiner observed the patient’s eye movements for nystagmus, which is normally absent [26]. A peripheral etiology was related to horizontal or horizontal-rotatory nystagmus, indicative of a vestibulo-ocular pathway abnormality.

#### 2.2.2. Paraclinical Vestibular Explorations

The vestibulo-ocular reflex was evaluated using the video head impulse test (VHIT) (VHIT Ulmer II, SYNAPSYS, Marseille, France) and videonystagmography (VNG), including kinetic tests and bithermal caloric tests (VNG Ulmer, SYNAPSYS). Each test corresponds to a specific vestibular frequency evaluation: VHIT for high, kinetic tests for middle and bithermal caloric tests for low frequencies. For more information, please refer to Appendix A [27,28,29,30]. Briefly, VHIT evaluates the patient’s eye movements when the head is moved abruptly by the physician in each semicircular canal (SCC) direction. If the result of this test was normal, the eyes stayed focused on the target (no movement) while the head moved, showing a normal vestibulo-ocular reflex gain (between 0.8 and 1.2) [28]. For the kinetic tests, the patient sat on a rotatory chair that moved in a sinusoidal way (sinusoidal harmonic acceleration test) or in quick accelerations followed by quick decelerations (impulsive rotational test). Response to the kinetic tests was defined as abnormal if there was a directional preponderance (normally absent) of the nystagmus cumulated curve of 2°/second or more [29]. Bithermal caloric tests were performed by irrigating the ear canals alternatively for 30 s with cold (30 °C) then hot (44 °C) water. An abnormal test response (indicative of canal paresis) was defined as a difference of 25% or more between the induced responses of the two ears [30].

#### 2.2.3. Definition of Vestibular Syndrome

Vestibular syndrome in cases of significant pathological vestibular function was diagnosed, clinically, as nystagmus triggered by the HST, associated or not with a Fukuda test of over 45°, and/or, paraclinically, as an abnormal VHIT, abnormal kinetic test or abnormal bithermal caloric test. Given its poor sensitivity and specificity [31] an isolated abnormal Fukuda test was considered insufficient to fulfill the criteria for vestibular syndrome.

### 2.3. Auditory Investigation

Auditory testing was performed with a Madsen Astera2 audiometer (Otometrics–Natus, Massy, France). The pure-tone average (PTA) was calculated using the pure-tone audiometry threshold for four frequencies (0.5, 1, 2, 4 kHz), according to the modified 1995 AAO-HNS criteria [32]. Hearing loss was defined as a PTA of 20 dB or more in at least one of these frequencies.

### 2.4. Treatment and Follow-Up

The efficacy of treatments prescribed was monitored in a purely descriptive retrospective way. Drugs used included betahistine (an analog of histamine that acts on the neuronal systems involved in recovery after vestibular damage [33]) and acetyl-DL-leucine (an acetylated derivative of the natural essential amino acid, which improves the central vestibular compensation during acute vertigo [34]).

### 2.5. Statistical Analyses

Statistical analyses were performed using GraphPad Prism 5.2 (GraphPad software, Prism, San Diego, CA, USA). To compare quantitative variables, the Mann–Whitney test was performed, whereas a Fisher exact test was used to compare qualitative variables. A *P*-value < 0.05 was considered significant.

## 3. Results

### 3.1. Clinical and Paraclinical Ear, Nose and Throat Investigations

#### 3.1.1. Characteristics of Patients Referred for Investigations

Among the cohort of 90 GS patients, 20 (22%) complained of vertigo. Four of them refused investigations, but 16 patients (10 women, median age 41 (IQR 38–47) years) were referred for neuro-otological investigations. Clinical characteristics are summarized in Table 1. Their follow-up period for GS ranged from 7 to 30 years (median 17.5 years) (Appendix A). These patients had low blood pressure in sitting and standing positions (a known symptom of GS) but no orthostatic hypotension. As expected in GS patients, they had markedly decreased potassium and magnesium plasma concentrations (2.9 ± 0.55 and 0.58 ± 0.17 mmol/L, respectively). In eight patients (50%), GS was complicated by chondrocalcinosis. The survey results showed that six patients (38%) reported that vertigo symptoms appeared when they were between the ages of 30 and 50. Vertigo was of short duration (less than 1 min) for 11 patients (69%), lasted for less than 20 min for two patients (13%) and for more than a day for two patients (13%) (missing data for patient 5). Vertigo was triggered by head rotation in eight patients (50%). Two patients (13%) described headaches that were not closely related to vertigo crises and did not fulfill the criteria for vestibular migraine [24] (Appendix A).

#### 3.1.2. Neuro-Otological Clinical Exam

Results of clinical neuro-otological examinations are summarized in Table 2.

All patients had bilateral normal tympanic membranes. None had cerebellar syndrome, gaze nystagmus or stance or gait abnormalities. None of the patients had vertigo triggered by the Dix–Hallpike maneuver, excluding benign paroxysmal positional vertigo. Six (38%) of the 16 patients had abnormal auditory tests. Four presented with bilateral hearing loss at high frequencies (mean hearing loss at 4 kHz was 33.75 ± 6.29 dB, mean age 45.5 years) and three with unilateral hearing loss at low frequencies (mean hearing loss at 500 Hz was 25 ± 5 dB); one patient had both (Table 2). Eight patients (50%) had abnormal clinical vestibular evaluations: Seven patients (44%) had postural deviations on the Fukuda test, meaning that there was a vestibulospinal pathway abnormality, and five (31%) had horizontal nystagmus triggered by the HST, meaning there was an abnormal vestibulo-ocular pathway.

#### 3.1.3. Paraclinical Vestibular Tests

VHITs were normal for all patients. Six (38%) of the 16 had significant abnormalities on the vestibular tests: kinetic vestibular tests were abnormal for four patients (25%) (Figure 1), and caloric tests were significantly abnormal for three patients (19%) (Figure 2). For one patient, both were abnormal.

#### 3.1.4. Diagnosis of Vestibular Syndrome

Seven (44%) of the 16 patients had vestibular syndrome. Abnormalities are summarized in Figure 3. None of these patients had a Menière’s disease triad.

### 3.2. Comparative Results

There were no significant differences regarding blood pressure, sodium and magnesium plasma levels, chondrocalcinosis history, hearing loss, gender or age between patients who had abnormal vestibular tests and those who had normal tests. Significantly, the increase in heart rate after change from sitting to standing position was similar between groups (9 ± 11 vs. 12 ± 7 bpm; *p* = 0.22). These comparisons are summarized in Appendix A.

### 3.3. Treatment

Based on the vestibular test results, two patients (13%) with vestibular syndrome (patients 11 and 15) were treated with betahistine (24 mg × 2 per day) with good results; both the intensity and frequency of the vertigo decreased during 2 years of follow-up. Two patients (patients 2 and 8) who complained of debilitating vertigo but who did not have vestibular syndrome or tinnitus also dramatically improved upon treatment with betahistine. Two patients (patients 1 and 2) were treated with acetyl-DL-leucine for acute vertigo crisis, but the treatment was not effective (Table 2).

## 4. Discussion

In GS, vertigo is often attributed to orthostatic hypotension. In this observational study, we reported on vertigo investigations in 16 patients without orthostatic hypotension, according to the consensus definition, that constituted 18% of our cohort of patients [10]. Self-evaluations revealed that the transition from lying to standing triggered the crisis in less than 50% of patients. This and other characteristics, including the finding that short duration rotatory vertigo could be triggered by rotation of the head and that vertigo could occur while lying down, were highly suggestive of a dysfunction of the inner ear. For this reason, we performed high-precision neuro-otological investigations in 16 adults of our cohort who complained of vertigo. We documented abnormal vestibular function in seven (44%) of them, highlighting, for the first time, the presence of a vestibular dysfunction in GS independent of orthostatic hypotension.

The two most frequent vestibular causes of vertigo are benign paroxysmal positional vertigo and Menière’s disease [35]. Benign paroxysmal positional vertigo is triggered by sudden movements of the head, relative to gravity, which induce migration of free-floating particles of calcium carbonate crystals in the semicircular canal [36]. Because GS patients are prone to develop chondrocalcinosis, likely related to magnesium depletion, we wondered whether calcium pyrophosphate crystals might be present in the inner ears of these subjects. We found no clinical evidence for benign paroxysmal positional vertigo in our population, however. In addition, there was no correlation between vertigo and chondrocalcinosis history. Moreover, the pathophysiology of pyrophosphate crystal deposits and calcium carbonate crystals differ. The pyrophosphate crystal deposits in GS are favored by magnesium depletion, which inhibits enzymatic degradation of pyrophosphate crystals, and, to our knowledge, these crystals have never been reported in the inner ear [37]. These results argue against benign paroxysmal positional vertigo in GS patients.

The symptoms presented by the patients overlap with those of Menière’s disease without corresponding to its definition. Menière’s disease is clinically defined by episodic vestibular symptoms associated with fluctuating low frequency hearing loss or aural symptoms (aural fullness or tinnitus) occurring over periods that range from 20 min to 12 h. Typically, subjects with Menière’s disease have normal VHITs, whereas the kinetic and caloric tests are often abnormal [23]. Among the seven GS patients with proven vestibular syndrome, five reported tinnitus with unilateral aural fullness or hearing loss, but none had a true Menière’s disease triad. Indeed, the duration of vertigo was shorter in our cohort (less than 20 min in 87% of cases) than is typical for Menière’s disease patients who have crises that typically last at least 20 min [23]. Another point is that vertigo onset occurred before 30 years in 44% of cases in our cohort, thus, earlier than usual in Menière’s disease, where the age of symptom onset typically ranges between 30 and 70 years [23]. However, this result can match with the early onset of symptoms in genetic Menière’s disease [38].

The pathophysiology of Menière’s disease is still unclear, but it is likely associated with an endolymphatic hydrops, which is thought to result from a dysregulation of ion transport in the endolymphatic sac [27]. The endolymphatic sac cells play crucial roles in sodium transport and in endolymph volume regulation. These cells express several transporters of sodium including NCC, the cotransporter that is deficient in GS patients [39]. We propose that in patients with GS, lack of NCC changes the composition and/or the volume of the endolymph compartment and that the renin–angiotensin system, activated by the renal loss of sodium, stimulates other salt transporters expressed by cells of the endolymphatic sac to partly counterbalance the defect. This could explain the variability in phenotype we observed in our cohort. Moreover, NCC mutation in the endolymphatic sac could lead to an accumulation of sodium in the endolymph. We hypothesize that sodium concentration remains normal in the GS endolymphatic sac due to the presence of aquaporins, which will counterbalance by retaining water with sodium, preventing an increase in sodium concentration in the endolymphatic sac but resulting in a hydrops in the endolymphatic sac.

To treat vertigo, four patients in our cohort (two of whom were diagnosed with vestibular syndrome) were treated long term with betahistine; all four patients reported improvement in the vertigo. In contrast, acetyl-DL-leucine, given for acute treatment during vertigo crises for two patients, did not improve the symptoms. Because of the small size of our population, we cannot definitively conclude whether or not either of these treatments will be broadly effective, and further studies are needed.

We must acknowledge that our study has the limitations of a retrospective analysis of patients affected by a rare disease: Among the 20 GS patients who reported vertigo out a cohort of 90 patients, only 16 were evaluated by an ear, nose and throat specialist. Our observations should thus be confirmed in a larger prospective study with systematic vestibular explorations with control populations, such as GS patients with orthostatic hypotension and/or control patients with vertigo but without GS. Moreover, systematic magnetic resonance imaging should be performed on GS patients with a proven vestibular syndrome to better understand the underlying physiopathology. Finally, we did not evaluate vestibular-evoked myogenic potentials to specifically study the functions of the otolith (saccule and utricle) and did not perform electrocochlearography to test auditory manifestations of the endolymphatic hydrops. These tests are usually performed when other vestibular tests are normal and would perhaps have shown vestibular abnormalities in patients who had normal results in the tests performed [40].

Our study does have strengths. Our population was well genotyped. Thus, the link between the NCC defect and vestibular dysfunction was strongly suggested. Furthermore, investigations were centralized in a department with expertise in vestibular investigations. This allowed us to document vestibular abnormalities with high prevalence, excluding chance, and strong objective criteria were used to define vertigo and vestibular syndrome that were documented in 7 out of 90 patients (8%).

In conclusion, we described, for the first time, abnormal vestibular function in patients diagnosed with GS. Our data suggest an important role of NCC in the regulation of endolymphatic composition and/or volume. Vestibular syndrome should be considered as a new extrarenal manifestation of GS, and nephrologists and physicians who treat these patients should be aware of this disabling manifestation, which requires specific evaluation and which can be treated.

## Figures and Tables

**Figure 1 jcm-09-03790-f001:**
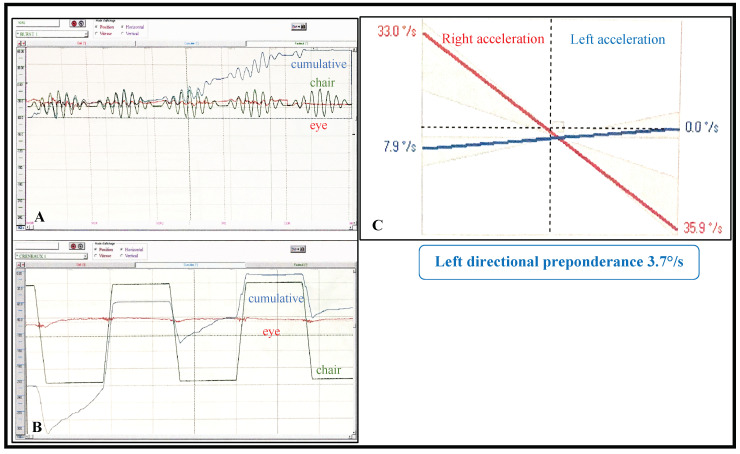
Results of a representative pathological kinetic test (patient 13). For method, see Appendix A. (**A**) Sinusoidal harmonic acceleration test result. Green curve indicates rotatory chair sinusoidal movements; red curve indicates eye movement (nystagmus); blue curve is the cumulative result of movements of both eyes. The blue curve shows a more intense nystagmus when chair is rotated to the left side: the blue ascendant curve represents a left directional preponderance. (**B**) Impulsive rotational test. The three curves (green, red and blue) represent results of the same item but triggered by different stimuli: the rotatory chair moves with quick acceleration in one direction, is then stabilized and then moves with quick deceleration in the opposite direction, the so-called “creneau” stimulation. The blue ascendant curve shows a left directional preponderance. (**C**) Summary of impulsive rotational test results showing a significant left directional preponderance of over 2° per second.

**Figure 2 jcm-09-03790-f002:**
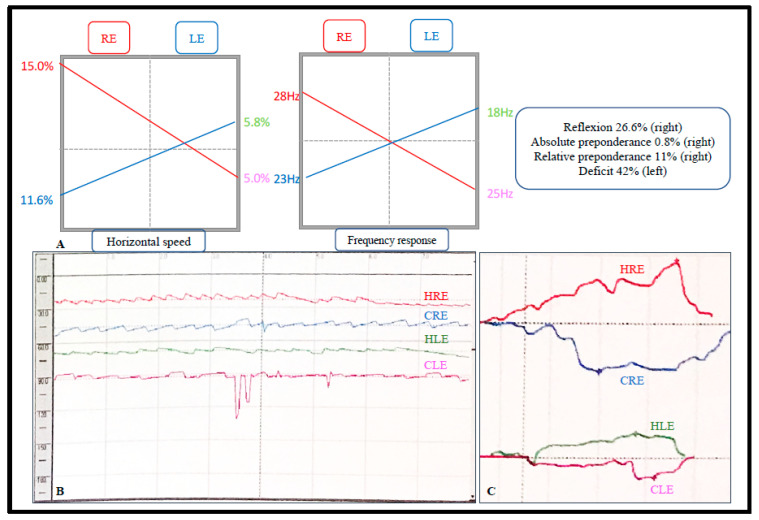
Results of a representative pathological caloric test (patient 6). For method, see Appendix A. (**A**) Graph summarizing the results of the caloric test, showing a left vestibule deficit of 42%. RE, right ear; LE, left ear. (**B**) Graph showing the nystagmus after stimulating (hot-water irrigation) or inhibiting (cold-water irrigation) each vestibule. SN, spontaneous nystagmus (dark line: spontaneous eye movement (no nystagmus)); HRE, hot water, right ear (red line: nystagmus after right-ear hot-water irrigation); CRE, cold water, right ear (blue line: nystagmus after right-ear cold-water irrigation); HLE, hot water, left ear (green line: nystagmus after left-ear hot-water irrigation); CLE, cold water, left ear (pink line: nystagmus after left-ear cold-water irrigation). (**C**) Comparison of nystagmus after irrigating the right ear (red and blue curves) and the left ear (green and pink curves). The intensity is weaker after irrigation of the left ear than of the right ear.

**Figure 3 jcm-09-03790-f003:**
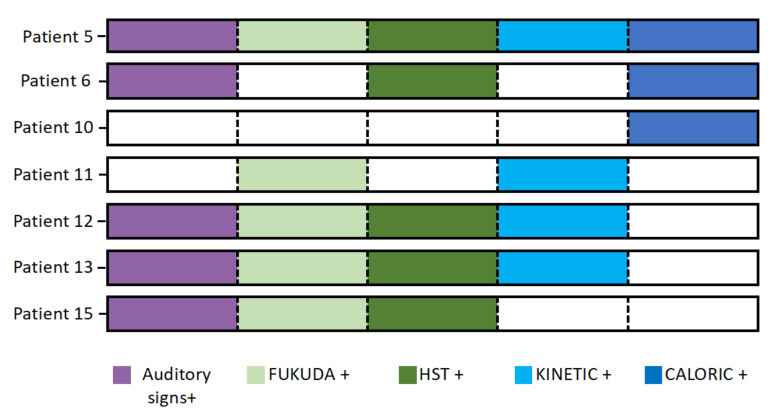
Association of clinical and paraclinical vestibular signs among the 7 patients with vestibular syndrome. Auditory signs: tinnitus, aural fullness or hearing loss; Fukuda +: positive Fukuda test; HST +: nystagmus triggered by head shaking test; kinetic +: kinetic test showing a directional preponderance over 2%; caloric +: caloric test showing a difference of 25% or more between the response of the two ears.

**Table 1 jcm-09-03790-t001:** Clinical characteristics of the patients.

Patient	Age	Sex ^a^	[K^+^] ^b^ mM [3.50–4.50]	[Mg^2+^] mM [0.64–0.90] ^b^	CCA ^c^	SBP/DBP (HR) ^f^ Standing Seated		Associated Symptoms ^g^	Triggers ^h^	Vertigo Duration	Frequency ^i^	Age of First Symptoms
1	40	M	2.2	0.65	+ ^d^	137/88 (76)	142/91 (93)	a	a,b	>1 day	c	30–50 y
2	38	F	3.1	0.60	+ ^d^	109/70 (88)	109/74 (100)	a,c,d	a	>1 day	a	30–50 y
3	46	M	3.1	0.82	-	118/66 (74)	119/73 (85)	-	b	<1 min	c	20–30 y
4	41	F	3.6	0.80	+ ^e^	114/73 (87)	101/73 (107)	-	c	<1 min	c	20–30 y
5	51	M	2.9	0.53	-	103/57 (77)	103/61 (79)	d	-	-	c	>50 y
6	47	F	3.3	0.52	+ ^e^	125/79 (86)	125/79 (86)	d	a,b	<1 min	a	30–50 y
7	41	F	2.2	0.53	+ ^e^	108/67 (81)	100/67 (88)	-	a,b	<1 min	c	20–30 y
8	44	M	2.6	0.40	+ ^d^	114/76 (61)	117/76 (78)	c	a	<1 min	b	>50 y
9	41	F	2.9	0.53	-	94/64 (80)	95/70 (86)	c	c	<20 min	c	30–50 y
10	28	F	3.2	0.70	-	117/72 (100)	109/76 (117)	-	b	<1 min	c	<20 y
11	37	F	3.4	0.78	-	124/79 (80)	132/82 (89)	a,b,c	a	<1 min	a	20–30 y
12	31	F	3.5	0.70	-	127/71 (84)	123/76 (99)	c,d	a	<1 min	a	<20 y
13	47	F	2.7	0.61	+ ^d^	106/77 (107)	97/77 (119)	-	c	<1 min	a	30–50 y
14	54	F	2.9	0.50	+ ^e^	103/64 (60)	98/80 (70)	a,b,c,d	b	<20 min	b	>50 y
15	53	M	2.7	0.49	-	116/65 (107)	110/69 (111)	d	a,b	<1 min	b	<20 y
16	34	M	2.6	0.57	-	119/81 (73)	123/79 (94)	-	c	<1 min	c	30–50 y

^a^ Sex: M, male; F, female. **^b^** Plasma concentrations. ^c^ CCA: X-ray documented chondrocalcinosis. ^d^ Infraclinical chondrocalcinosis. ^e^ Symptomatic chondrocalcinosis. ^f^ SBP/DBP (HR): systolic/diastolic blood pressure in mmHg (heart rate in bpm). ^g^ Associated symptoms: a, nausea; b, headache; c, history of rotatory vertigo; d, tinnitus. ^h^ Triggers: a, head rotation; b, orthostatic; c, spontaneous. ^i^ Frequency: a, permanent or at least once a week; b, at least once a month; c, less than once a month.

**Table 2 jcm-09-03790-t002:** Neuro-otological clinical exams and paraclinical test results for Gitelman syndrome patients with vertigo.

Patient	Aural Signs ^a^	Hearing Loss ^b^	Fukuda Deviation	HST ^c^	VHIT ^d^	Kinetic ^e^	Caloric ^f^	Vestibular Syndrome	Drug ^g^	Drug Efficacy
1		25 dB 4000 Hz	-	-	-	-	-	-	+(a)	-
2	a	-	-	-	-	-	-	-	+(a,b)	+(b)/−(a)
3		-	-	-	-	-	-	-	-	-
4		-	-	-	-	-	-	-	-	-
**5**	**b**	**35 dB 4000 Hz** **20 dB 8000 Hz**	**+**	**nystagmus**	**-**	**3.2%**	**30%**	**+**	**-**	**-**
**6**	**b**	**25 dB 500 Hz 40 dB 4000 Hz** **60 dB 8000 Hz left** **35 dB 8000 Hz right**	**-**	**nystagmus**	**-**	**-**	**42%**	**+**	**-**	**-**
7		-	-	-	-	-	-	-	-	-
8		35 dB 4000 Hz	-	-	-	-	-	-	+(b)	+
9		-	-	-	-	-	-	-	-	-
**10**		-	-	-	-	-	**43%**	**+**	**-**	**-**
**11**		-	**+**	**-**	**-**	**3.5%**	**-**	**+**	**+(b)**	**+**
**12**	**a**	-	**+**	**nystagmus**	**-**	**2.3%**	**-**	**+**	**-**	**-**
**13**		**20 dB 150 Hz**	**+**	**nystagmus**	**-**	**3.7%**	**-**	**+**	**-**	-
**14**	b	30 dB 500 Hz right	+	-	-	-	-	-	-	-
**15**	**a**	-	**+**	**nystagmus**	**-**	**-**	**-**	**+**	**+(b)**	**+**
16		-	+	-	-	-	-	-	-	-
Total n (%)	6 (37.5)	6 (37.5)	7 (44)	5 (31.2)	0 (0)	4 (25)	5 (31.2)	7 (44)	5 (31.2)	4 (25)

Text is in bold for patients with vestibular syndrome. ^a^ Aural signs: a, tinnitus or b, tinnitus associated with aural fullness. ^b^ Hearing loss at high or low frequency or both. ^c^ HST: head shaking test. ^d^ VHIT: video head impulse test. ^e^ Abnormal if > 2%. ^f^ Abnormal if > 25%. ^g^ Drug: a, acetyl-DL-leucine or b, betahistine.

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
