# Peer review of "Investigation of Vestibular Function in Adult Patients with Gitelman Syndrome: Results of an Observational Study"

_jcm, 2020, doi:10.3390/jcm9113790_

Round 1
Reviewer 1 Report
The manuscript reports vestibular dysfunction and auditory signs and symptoms in a series of adult patients with Gitelman's syndrome (GS). Twenty out of 90 patients with GS complained of vertigo apparently not caused by orthostatic hypotension. Sixteen were investigated.
The study draws attention to a clinical manifestation not usually considered in the study of individuals with GS.
There are important methodological limitations, which are mentioned by the authors to a great extent. Mostly, the absence of appropriate controls for comparison.
Suggestions to improve the manuscript:
- Were all 90 patients systematically asked at diagnosis or follow-up if they had vertigo or hearing symptoms? That is, did the other 70 patients not spontaneously complain or explicitly state that they did not have these symptoms.
- The distribution of patients indicated in the last statements of the abstract is somewhat confusing for the general reader. Please, clarify.
- Some information from the literature mentioning the percentage of patients with GS who complain of vertigo would be of interest.
- A Table showing some additional information on the patients, such as age at diagnosis, follow-up period, gene mutations, received treatment, if any, for GS, etc., would be useful.
- Do the authors have data from paediatric patients with GS? Could the authors comment on that?
Reviewer 2 Report
The manuscript by Drs Alexandru, et al, describes their work with patients having Gitelman's syndrome. The patient population is well established genetically.
They report that 22% of their patients reported having vertigo. They then carried out a very thorough investigation of their inner ear function and conclude that the dysfunction they found could be attributable to the transport defect of Gitelman's syndrome.
This is an observational study but demonstrates an interesting aspect of Gitelman's syndrome that is not well appreciated.
Reviewer 3 Report
This is an interesting study about the vestibular phenotype of SLC12A3 mutation which causes Gitelman syndrome. So far, there is no evidence of hearing and vestibular phenotype in NCC dysfunction. Although the paper is well written and organized, there are several issues that should be addressed for publication.
1. The parameters for vestibular function test should be modified or descirbed more in detail. Authors used parameters for abnormal vestibular function as follows; presence of corrective saccade for vHIT, directional preponderance for kinetic test. Why didn't they use gain and corrective saccade for vHIT and gain/phase/asymmetry/time constant for rotary chair test? Those parameters are universally accepted for the interpretation of those tests. It would be better if the authors used the parameters in those tests. Since the SLC12A3 dysfunction can cause bilateral vestibular dysfunction (mild or severe) according to the authors hypothesis, those parameters seems to be better for representing the vestibular abnormalities. Mild bilateral vestibular function cannot be detected in bithermal caloric test.
2. In table 1, it would be better if the authors add reference values for [K+] and [Mg2+].
3. In table 1, three patients had vertigo triggered only by orthostatic position change. Two of them did not have abnormality in neuro-otological exam and paraclinical test. Did the authors think those patients have real vestibular phenotype?
4. Is there any patients who showed hearing loss higher than 4000Hz?
5. In Figure 2, I could not find abbreviation SN in the figure, although it appeared in figure legend. Please insert it in the figure.
6. I think treatment using betahistine and acetyl-DL-leucine for vertigo treatment can be removed from the manuscript. There is no scientific evidence or background for using those medications in treating the vestibular symptom of Gitelman syndrome.
7. In Discussion, line 17 - 19, authors described that there was no correlation between vertigo and chondrocalcinosis history. I could not find any analysis about this in the result. Was there no difference in the proportion of chondrocalcinosis history between the patients with vertigo and others?
8. In Discussion, line 27, the duration of vertigo in Meniere's disease should be changed from "20 min to 24 hours" to "20 min to 12 hours" according to the Diagnostic criteria of 2015 Barany Society.
9. In Discussion, line 32 - 34, the vertigo onset in the patients with genetic Meniere's disease can be earlier than the patients with sporadic Meniere's disease, especially in patients with definite mutation of genes.
10. Did the authors think that mutation of SLC12A3 in the endolymphatic sac can cause endolymphatic hydrops or other vestibular dysfunction by elevating [Na+] in the inner ear. It would be better for understanding the vestibular phenotype if the authors describe their hypothesis in detail in Discussion.
11. In Table S1, I found the duration of vertigo for several hours in 13% of the patients; however, there was no patients with the vertigo duration in Table 1. Why was there discrepancy between Table 1 and Table S1?
12. In Table S1, it would be better if the authors include the questionnaire if lying down triggered vertigo in the patients, because vertigo in the position change frequently causes vertigo in PSCC BPPV. In addtion, there is no questionnaire for aural fullness as an associated signs for vertigo. Did the authors ask more detail during history taking? If so, it would better to describe about that in the manuscript.
